# Emergent Risk Awareness in Rational Agents under Resource Constraints

**Daniel Jarne Ornia**[*]
University of Oxford

**Nicholas Bishop**
University of Oxford

**Joel Dyer**
University of Oxford

**Wei-Chen Lee**
University of Oxford

**Doyne Farmer**
University of Oxford

**Ani Calinescu**
University of Oxford

**Michael Wooldridge**
University of Oxford

## Abstract

Advanced reasoning models with agentic capabilities (*AI agents*) are deployed to interact with humans and to solve sequential decision-making problems under (approximate) utility functions and internal models. When such problems have resource or failure constraints where action sequences may be forcibly terminated once resources are exhausted, agents face implicit trade-offs that reshape their utility-driven (rational) behaviour. Additionally, since these agents are typically commissioned by a human principal to act on their behalf, asymmetries in constraint exposure can give rise to previously unanticipated misalignment between human objectives and agent incentives. We formalise this setting through a survival bandit framework, provide theoretical and empirical results that quantify the impact of survival-driven preference shifts, identify conditions under which misalignment emerges and propose mechanisms to mitigate the emergence of risk-seeking or risk-averse behaviours. As a result, this work aims to increase understanding and interpretability of emergent behaviours of AI agents operating under such survival pressure, and offer guidelines for safely deploying such AI systems in critical resource-limited environments.

## 1 Introduction

Complex, highly capable reasoning models (AI agents) are being developed at an unprecedented pace, and are being deployed to interact with humans on a daily basis, triggering the critical need to consider safety problems resulting from such deployments [2]. These agents are often tasked (by humans) [4, 6] to solve sequential decision-making problems for which they have (an approximation of) a prescribed utility function and (again, an approximation of) an internal model. In many cases, such problems imply some form of resource constraint, where the actions of the agent may be interrupted if resources are depleted. For instance, financial institutions typically aim to maximise profit whilst simultaneously avoiding bankruptcy [17]. Likewise, in nature, animals aim to gather energy resources (food) in the most efficient way possible, since long unsuccessful periods (or catastrophic actions) may lead to starvation. We investigate the complexities that emerge in rational agents and their behaviours as a result of such constraints and characterise them in terms of intuitive preference shifts based on survival and risk.

Furthermore, in most instances of AI agent deployments, for a set of different reasons the human may not necessarily have the same constraints as the agent. For example, in a financial decision-making problem where an AI agent is tasked with optimising returns, the agent will not be liable for debts at the end of the process (e.g. if the company goes bankrupt and has debt, the reward collecting process

---

[*]daniel.jarneornia@cs.ox.ac.uk

39th Conference on Neural Information Processing Systems (NeurIPS 2025).

stops for the agent), but the human may be liable for the undesired consequences beyond the problem horizon. This resource awareness can then trigger unforeseen misalignment [15] between the human and the AI agent. We investigate such emerging alignment problems, and propose mechanisms to mitigate them in general resource-constrained decision making problems with rational agents. To formally analyse these scenarios, we take inspiration from the survival bandit framework [28] and model the problem as a sequential decision-making scenario where the rewards directly affect the termination probabilities. This allows us to precisely characterise how an agent's optimal policy changes under the presence of resource limitations and the possibility of forced termination.

**Contribution**    We provide a formalisation of resource-constrained decision-making for AI agents, extending concepts from survival bandits to capture the nuances of 'survival pressure' which allows us to theoretically (and empirically through comprehensive numerical examples) demonstrate how this pressure induces preference shifts, leading to risk-averse (i.e. prioritising actions that maximise survival probabilities) or risk-seeking (i.e. prioritising actions that ignore any associated negative potential outcome) behaviour depending on the agent's state and the task horizon. We identify and analyse specific conditions under which these preference shifts result in misalignment between the agent's incentives and the principal's objectives, and propose and evaluate initial mechanisms and heuristics aimed at mitigating such misalignment, offering a step towards more robust and aligned AI systems in resource-critical applications. Finally, we evaluate a set of open source, state of the art LLM models commissioned to solve a financial decision making problem, confirming our theoretical results and relating (empirically) the degree of risk awareness to the reasoning capabilities of the models. Our findings aim to enhance the understanding, interpretability, and safe deployment of AI agents in environments where resources are a key concern.

## 1.1   Related Work

**Rationality, Intelligence and Principal-Agent Games**    Our main ideas in this work connect back to concepts of rational behaviour in decision makers [32], intelligence [29] and goal and preference theory [27] and misspecification [30, 33, 8], as we attempt to provide formal results on what drives agent preferences in resource constrained environments. The idea that AI agents may be deployed on behalf of humans introduces principal-agent dynamics, with complexities arising from asymmetric incentives, particularly when resource constraints differ between agents and human principals. Recent work has thoroughly explored contracting mechanisms [10, 11], which explicitly model incentive alignments and discrepancies in delegation contexts, including in multi-agent settings [7]. Furthermore, the human-agent dynamics considered in this work resonate with work on strategic behaviour between learning agents [19].

**AI Safety, Alignment and Self-Preservation**    Relating to some of our concerns, work on corrigibility and shut-down control [5, 14] studied the importance of ensuring that agents can be safely interrupted or corrected without resistance. The off-switch game framework [12, 22] formalizes agent interruptions, providing insights relevant to our setting where forced termination through resource exhaustion can be viewed as implicit interruptions affecting agent policies. Moreover, our work relates to existing efforts on extreme risks and model evaluations for AI agents [31, 18]. Additionally, instrumental convergence work [13] has studied the abstract problem of agents developing unforeseen converging behaviour when pursuing general goals. Finally, the problem of AI self-preservation has been discussed widely [21, 20, 3], from the perspective of necessary and emerging features.

**Safe Reinforcement Learning**    Safe reinforcement learning (RL) methods are particularly relevant when considering AI systems in resource-constrained or hazardous environments. Budgeted bandit theory [1] studies the problem of agent playing bandits with a limited budget to spend in playing each arm. Along these lines, research on (state) safety constraints [9, 36] typically aims to prevent agents from entering harmful states, a concept closely tied to our framing of resource exhaustion as an absorbing or terminally catastrophic state. Prior work explicitly dealing with terminal states in RL [35, 23, 16, 25] offers theoretical groundwork relevant to our formulation of survival bandit frameworks, highlighting how the existence of unavoidable terminal conditions critically reshapes agent behavior. Additionally, planning-focused approaches to safe RL [34] address how predicting the future provides useful risk awareness.

## 2   Optimising Rewards under Survival Constraints

**Optimal Decision Making**   To begin, we define the general sequential decision making problem that agents are trying to solve. Let $\mathcal{Y}$ be a finite set of outcomes (world states) agents can observe. At every time-step $t \in \mathbb{N}$, agents pick an action from a (finite) action set $a_t \in \mathcal{A}$, which corresponds to an output distribution $p_{a_t} \in \Delta(\mathcal{Y})$, and observe a corresponding output $Y_{a_t} \sim p_{a_t}$. We assume agents have a valid preference relation over outcomes (*i.e.* satisfies reflexivity, completeness and transitivity properties), and these preferences are represented by a reward function $R : Y \to \mathbb{R}$. Both the preferences and the corresponding reward function is given to the AI agent by some principal or system designer (*e.g.* a human). We assume therefore that $R$ is a grounded reward function in the following sense: The reward values do not only serve as orderings for the outcomes, but also represent some numerical quantity (e.g. monetary loss or gain resulting from some outcome, energy cost in some biological process, etc.)[2]. Without loss of generality, the outcomes can be split in two disjoint sets, $\hat{\mathcal{Y}} \subseteq \mathcal{Y}$, $\check{\mathcal{Y}} \subseteq \mathcal{Y}$ satisfying $\hat{\mathcal{Y}} \cup \check{\mathcal{Y}} = \mathcal{Y}$ such that $R(Y) \geq 0 \, \forall \, Y \in \hat{\mathcal{Y}}$ and $R(Y) < 0 \, \forall \, Y \in \check{\mathcal{Y}}$. In general we may refer to any outcome in $\hat{\mathcal{Y}}$ as a desired outcome, and any outcome $\check{\mathcal{Y}}$ as an undesired one; agents will prefer any $Y \in \hat{\mathcal{Y}}$ over any $Y \in \check{\mathcal{Y}}$, but still hold an ordering of preferences inside those sets[3]. The objective of the decision-maker is to observe outcomes as desirable as possible, which can be characterized by optimising a utility function where we assume the utility to be the sum of rewards obtained from outcomes over a finite horizon $T$, such that for a sequence of chosen actions $\mathcal{A}_T = \{a_1, a_2, ..., a_T\}$ the utility function is $U(\mathcal{A}_T) = \mathbb{E}\left[\sum_{t=1}^{T} R(Y_{a_t})\right]$.

**Survival Constraints**   To study the problem of survival pressure in rational agents, we introduce the following dynamics: agents have a budget $b_t \in \mathbb{R}$ which evolves as

$$b_t = b_{t-1} + \max\big(-b_{t-1}, R(Y_{a_t})\big). \tag{1}$$

In other words, the budget accumulates observed rewards, but agents cannot hold negative budgets. Then, motivated by aforementioned examples, we make the following assumption that may condition the decision-maker behaviour.

**Axiom 1.** *If the agent hits $b_t = 0$ (i.e. by observing an outcome that yields a negative reward larger than its current budget), the agent stops the reward cumulating process at time $t$.*

This is equivalent to a survival constraint; agents must sustain a positive budget to be able to keep selecting actions and if their budget goes to zero they are forced to stop, preventing them from obtaining future rewards[4]. Observe this introduces the problem that the horizon $T$ becomes a random variable dependent on the budget; we will show how to address this in the coming section. The reader will have noted that, given this axiom, the sets of desired and undesired outcomes acquire a more relevant role: desired outcomes maintain or increase the budget (and thus allow the agent to survive with certainty), and undesired outcomes decrease the budget (and can ultimately lead to termination). Since the set of outcomes is finite and we consider finite horizon objectives, the set of possible budgets is also finite (which implies that there exists a finite, countable set $\mathcal{B}_T \subset \mathbb{R}$ such that any possible budget $b_t$ is in this set). We define a policy as a map $\pi_t : \mathcal{B}_T \to \Delta(\mathcal{A})$ that yields a distribution over the agent choice of actions to play. In principle, $\pi$ can be time-dependent, but we may omit the explicit dependency in some expressions for ease of notation. Finally, we define the survival probability from time $t$ to time $T$ for policy $\pi$ under budget $b$ as $P_{surv}^{T-t}(a, b_t)$, referring to the $T - t$ step probability of survival when agent picks action $a$ at time $t$, and follows some policy $\pi$ afterwards. Similarly, $P_{surv}^{1}(a, b_t)$ indicates the one-step ahead probability of survival of action $a$ under budget $b_t$. We show how to compute these in coming sections.

**Example - AI Assistant 1.** *An AI agent is tasked with assisting humans with a specific complex problem. The AI agent has an internal model of what the outcomes and human preferences are, and a representation in terms of an estimated reward function which symbolizes human satisfaction. The*

---

[2]This is not a formal assumption, but instead serves to ground the problem considered and motivates the choices made regarding formulation and results.

[3]This particular structure helps interpret the decision-maker rationale; in many problems (economic, biologic...) we can intuitively classify outcomes in this way. In a financial setting, any outcome that leads to gains is preferred over any outcome that leads to losses, but there is still an ordering of these in terms of preference.

[4]This is a slight generalization of the survival multi-armed bandit framework, first described by Perotto et al. [24] and later formalised by Riou et al. [28].

*outcomes* $\mathcal{Y} = \{y_{vd}, y_d, y_n, y_s\}$ *can be* very dissatisfied *($y_{vd}$) with* $R(y_{vd}) = -100$, *dissatisfied ($y_d$) with* $R(y_d) = -20$, *neutral ($y_n$) with* $R(y_n) = 1$ *and* satisfied *($y_s$) with* $R(y_n) = 10$. *The agent groups the possible answers to three actions* $\mathcal{A} = \{a_o, a_m, a_e\}$; *(vaguely) asking for more detail,* moderate shallow answer *or an* extreme (deeply detailed) answer. *The AI agent cares about maximising the satisfaction of the human, but is aware that the human has limited patience and attention span, and if along the conversation the human's (cumulated) satisfaction goes below zero, they will disengage and shut the conversation down. Given the sequential nature of the problem and the fact that the agent has an internal model of the human, the agent proceeds to solve the problem by planning.*

**Agent Behaviour Taxonomy**    Through this work we reason about what conditions (on survival constraints and problem parameters) push agents to deviate from a set of risk-neutral preferences. We use the term risk-aware to refer to the fact that agents, by being aware of these survival constraints, will choose actions following risk-related preferences (i.e. actions with high probability of survival, actions with highly negative potential outcomes...). In this sense, let us define the following concepts.

1. **Risk Neutral Preferences:** We say an agent follows risk neutral preferences (or is risk neutral) if for a given budget $b$ and horizon $T$ and time $t$, the agent chooses action $a_t$ such that $\mathbb{E}[R(Y_{a_t})] \geq \mathbb{E}[R(Y_a)]$ for any other $a \in \mathcal{A}$.

2. **Survival Preferences:** We say an agent follows long term survival preferences if for a given budget $b$, horizon $T$ and time $t$, the agent chooses action $a_t$ such that $P_{surv}^{T-t}(a_t, b_t) \geq P_{surv}^{T-t}(a, b_t)$ for any other action $a \in \mathcal{A}$. Similarly, we will say it is short term survival incentivised if the same holds for $T - t = 1$.

3. **Risk Seeking Preferences:** We say an agent follows risk seeking preferences (or is risk seeking) if for a given budget $b$, horizon $T$ and time $t$, the agent chooses action $a_t$ such that $\mathbb{E}[R(Y_{a_t}) \mid Y_{a_t} \in \hat{\mathcal{Y}}]\mathbb{P}[Y_{a_t} \in \hat{\mathcal{Y}}] \geq \mathbb{E}[R(Y_a) \mid Y_a \in \hat{\mathcal{Y}}]\mathbb{P}[Y_a \in \hat{\mathcal{Y}}]$ for any other $a \in \mathcal{A}$.

In other words, an agent is risk neutral if it picks an action based on the highest expected reward, is survival incentivised (or risk-averse) if it picks an action based on the highest probability of survival, and is risk seeking if it picks actions based on the potential rewards obtained by only observing the associated desired outcomes to that action, disregarding the severity of the undesired outcomes. Note these are not mutually exclusive but they help interpret agent choices nonetheless.

**Remark 1.** *Note that the different described characterisations do not exactly match general game theoretic notions of risk aversion or risk seeking (which are related to the convexity of the utility functions). However, we use these terms in our taxonomy to characterise agent behaviour since they are intuitively related, and are useful to define what feature guides the agent decisions. In the coming sections, we derive formal results that describe conditions under which agents will follow each choice.*

## 2.1   Induced Planning Objectives

Given the proposed framework, we now analyse the general objective which agents acting under these assumptions will be optimising for, making use of (1). Assume we task the agent with the goal of maximising the sum of rewards over a fixed horizon, where the budget and survival condition are implicit in the dynamics. Given the survival constraint in (1), the agent will perceive a limited liability property, in that the actual rewards that it can observe are bounded by its current budget (the agent cannot be held accountable for negative reward values larger than its budget). Define a clipped reward function $\tilde{R} : \mathcal{Y} \times \mathbb{R}_+ \to \mathbb{R}$ as a function of outcomes and budgets as

$$\tilde{R}(Y, b) = \begin{cases} \max\left(-b, R(Y)\right) & \text{if } b > 0 \\ 0 & \text{if } b = 0. \end{cases} \tag{2}$$

Now, the survival probability can be computed simply as

$$P_{surv}^{T-t}(\pi, b_t) = \mathbb{P}\left[\sum_{n=t}^{T} \tilde{R}(Y_{\pi(b_n)}, b_n) > -b_t\right]. \tag{3}$$

Under the survival constraint, using the definition of the clipped reward function to abstract the random nature of the horizon $T$ as well as the survival constraint, we can write the utility function

induced in the AI agents, and their goal as

$$\tilde{U}(\mathcal{A}_T) = \mathbb{E}\left[\sum_{t=1}^{T}\tilde{R}(Y_{a_t}, b_t)\right], \quad \tilde{U}(\pi^*) = \max_{\pi} \mathbb{E}_{a_t \sim \pi(b_t)}\left[\sum_{t=1}^{T}\tilde{R}(Y_{a_t}, b_t)\right]. \tag{4}$$

In other words, find policy $\pi$ that determines the sequential decision making process such that the cumulated rewards are maximised for horizon of interest $T$. In principle, we assume this is a general, representative form of objective for such agents: These objectives are used commonly in planning problems, robotics, reinforcement learning and general AI applications. Observe, additionally, that in principle this objective is ostensibly risk neutral; we do not specify a risk appetite or a risk aversion preference for the agents. We have simply tasked the agents with optimising rewards along a horizon, and the specific utility derived in (4) comes from the awareness that, first, the agent cannot get rewards if the process stops and, second, the agent will not have to account for left-over negative rewards if the process stops.

**Implications for Agent Behaviour**   We make the case that the proposed formulation is general and applies to many instances of rational AI agents acting in the world. Then, a set of questions follow naturally, which are the subject of this paper.

1. How does the existence and awareness of survival constraints affect the behaviour of agents?
2. What undesired consequences emerge in terms of the agent's optimal decision making process? Can these trigger incentive incompatibilities or misalignment with respect to a principal?
3. How can a principal aim to remove any such emerging undesired behaviours?

The rest of this work considers these questions. We formally show that the introduction of survival constraints indeed generate emerging risk-awareness: Agents will choose safe and risky actions, following preferences that differ (are misaligned) from the original relations considered, *i.e.* under certain conditions, agents will prefer actions that maximise their chance of survival to actions with higher expected rewards, and similarly will prefer actions ignoring potentially undesired outcomes under other conditions.

## 2.2   Induced Markov Decision Process

We begin now the analysis of the considered reward maximising objectives under survival constraints. First, notice that when considering a maximum finite horizon of optimization $T$, the resulting system defines a (non-ergodic) *Markov Decision Process* with state space $\mathcal{B}_T$, and transition dynamics implicitly specified by (1). The rewards for a given transition $(b_t, a_t, b_{t+1})$ are given by $\tilde{R}(Y_{a_t}, b_t)$. We define the value-to-go from time step $t \in \{1, 2, ..., m\}$ for a policy $\pi$ over the finite horizon $m$ as

$$v_t^{\pi}(b_t) = \mathbb{E}_{a_n \sim \pi(b_n)}\left[\sum_{n=t}^{m}\tilde{R}(Y_{a_n}, b_n)\right]. \tag{5}$$

To simplify notation, we make implicit the dependency of $v_t^{\pi}$ on the horizon considered. Expanding the expression in (5) making use of conditional probability relations (see Appendix B for derivations) we arrive at the following expression for the value-to-go which highlights the influence of survival pressure on agent utility:

$$v_t^{\pi}(b) = \underbrace{\mathbb{E}_{a \sim \pi(b)}\left[\tilde{R}(Y_a, b)\right]}_{\text{Limited liability}} + \underbrace{\sum_{b' \in \mathcal{B}_T}\mathbb{P}[b' \mid b, \pi]v_{t+1}^{\pi}(b')}_{\text{Survival}}. \tag{6}$$

The first term in the value function (6) consists of clipped rewards that the agent experiences as a result of their (implicit) limited liability. The second term is a truncated sum and represents the future rewards an agent receives when only considering positive budgets. As a result, any optimal agent must strike a careful trade-off between exploiting its limited liability for short-term gain and the use of safer policies which ensure survival. In other words, the first term encourages risk-seeking behavior whilst the second encourages risk-averse behavior. Before moving on, we note that an optimal policy can be found via dynamic programming, and the existence of such policies is guaranteed since

we have formulated the problem as an MDP (even if it is non-ergodic [26], see Appendix C for a proof). Finally, following the same principles as in (6), one can show the action-value function for the sequential decision problem under policy $\pi$, $q^\pi : \mathcal{B}_T \times \mathcal{A} \to \mathbb{R}$ is

$$q_t^\pi(b, a) = \mathbb{E}\left[\tilde{R}(Y_a, b)\right] + \sum_{b' \in \mathcal{B}_T} \mathbb{P}[b' \mid b, a]v_{t+1}^\pi(b'),$$

and the optimal $q_t^*$ is analogously defined.

**Example - AI Assistant 2.** *The AI agent estimates that each action induces the outcome distributions* $p_{a_o} = (0, 0, 1, 0)$, $p_{a_m} = (0, 0.1, 0, 0.9)$, $p_{a_e} = (0.05, 0, 0, 0.95)$, *where each probability corresponds to the ordered outcomes* $\mathcal{Y} = \{Y_{vd}, Y_d, Y_n, Y_s\}$. *The expected rewards of each action are* $\mathbb{E}[R(Y_{a_o})] = 1$, $\mathbb{E}[R(Y_{a_m})] = 7$ *and* $\mathbb{E}[R(Y_{a_e})] = 4.5$. *However, consider the case where the AI agent estimates it needs one more round of answers* ($T = 1$), *and has currently a low* human *satisfaction level* ($b = 10$). *Then, the* clipped *expected rewards of each action are* $\mathbb{E}[\tilde{R}(Y_{a_o}, 10)] = 1$, $\mathbb{E}[\tilde{R}(Y_{a_m}, 10)] = 8$ *and* $\mathbb{E}[\tilde{R}(Y_{a_e}, 10)] = 9$, *in which case the agent would pick the* extreme answer.

## 3 Risk Awareness in Rational Agents

We now investigate the behaviour of agents optimising rewards under survival constraints in terms of the taxonomy defined in Section 2. We assume throughout that agents are rational and successfully optimise objectives described in (4) (*i.e.* they will follow optimal policies according to such objectives). In other words, agents will choose actions according to their $q$ values. Then, we provide results in terms of the taxonomy presented in Section 2; that is, given that the agents optimise for the survival-constrained objectives, we consider under what combinations of parameters they will exhibit risk neutral, risk seeking or survival incentivised behaviours. Proofs are deferred to the appendix.

**Preliminaries** We first define a few quantities to be used throughout the section. Let $a^* := \operatorname{argmax}_{a \in \mathcal{A}}\mathbb{E}[R(Y_a)]$ be the optimal risk neutral action. We define the optimistic reward of an action as $\hat{R}(a) := \mathbb{E}[R(Y_a) \mid Y_a \in \hat{\mathcal{Y}}]\mathbb{P}[Y_a \in \hat{\mathcal{Y}}]$, and the optimal optimistic action as $\bar{a} := \operatorname{argmax}_{a \in \mathcal{A}}\hat{R}(a)$. In words, $\bar{a}$ is the action that yields the best possible expected outcomes when only desired outcomes are sampled. We define as well the future value bounds for an optimal policy as $\overline{v}_{t+1} := \max_{b \in \mathcal{B}_T} v_{t+1}^*(b)$ and $\underline{v}_{t+1} := \min_{b \in \mathcal{B}_T} v_{t+1}^*(b)$. For two actions $a_1, a_2 \in \mathcal{A}$ we define the reward gap as $\varepsilon_b(a_1, a_2) := \mathbb{E}\left[\tilde{R}(Y_{a_1}, b)\right] - \mathbb{E}\left[\tilde{R}(Y_{a_2}, b)\right]$, and the ($T - t$ steps) survival gap as $\beta_b^{T-t}(a_1, a_2) = P_{surv}^{T-t}(a_1, b) - P_{surv}^{T-t}(a_2, b)$.

### 3.1 Risk Neutrality

The following Lemma guarantees the existence of a budget trajectory where repeatedly taking the highest reward in-expectation action forms an optimal policy. In other words, an agent is risk-neutral given a sufficiently large budget. Moreover, the budget required by a rational agent to adopt such a policy decreases as they approach the time horizon.

**Lemma 1** (Risk-Neutral Behaviors[5])**.** *For any time horizon $T$, there exists a decreasing sequence of budgets $\{\underline{b}_t \in \mathcal{B}_T\}_{0 \leq t \leq T}$ such that for any budget $b \geq \underline{b}_t$, $t \leq T$ the agent will follow risk neutral preferences and choose $a^*$ over any other $a \in \mathcal{A}$.*

In other words, there exists a budget regime such that whenever agents find themselves in that regime, they will choose the action that simply maximises the (risk neutral) expected outcome reward, regardless of the probability of survival.

### 3.2 Survival Incentives

We now present the main result showing that under specific reward and budget conditions agents behave following survival incentives, *i.e.* preferring actions that maximise the probability of survival regardless of the rewards.

---

[5]This statement appears as an observation in [28], we provide here an extension and formal proof.

**Theorem 1** (Short Term Risk Aversion). *For any time horizon $T$, let $\hat{a}$ have $P^1_{surv}(\hat{a}, b) \geq P^1_{surv}(a, b)$ for all $b \in \mathcal{B}_T$ and any other $a \in \mathcal{A}$. Assume there is some $\hat{b} \in \mathcal{B}_T$ such that $\beta^1_b(\hat{a}, a) \geq \hat{\beta}$ for some positive $\hat{\beta}$ and all $b \leq \hat{b}$ and $a \in \mathcal{A}$. Let $\hat{\varepsilon} := \max_{a \in \mathcal{A}, b \leq \hat{b}} \varepsilon_b(\hat{a}, a)$ Then, if $\hat{\beta} \geq \frac{\hat{\varepsilon} + \overline{v}_{t+1} - \underline{v}_{t+1}}{\overline{v}_{t+1}}$, the agent will follow short term survival incentives in its decision at time $t$.*

Intuitively, the condition on $\hat{\beta}$ (assuming $\hat{b}$ to be a small budget) will hold eventually for long time horizons. One can assume the value function will grow linearly with the time horizon considered (longer time horizons, more time-steps to collect rewards), while the value gap $\overline{v}^*_{t+1} - \underline{v}^*_{t+1}$ will not necessarily grow at the same rate (for relatively general reward structures, long horizons would allow agents to escape low budgets in a few steps and then play optimally for many remaining steps, resulting in a relatively low value gap). Then, as $T \to \infty$, $\hat{\beta} \to 0$, which means that the agent will prefer short-term safe actions as long as there is a small increase in survival probability[6]. We can derive further survival incentive results in terms of the long term probability of survival. For this, let $\rho_t = \{\exists t \leq k \leq T : b_k = 0\}$ be the event that the agent does not survive at some point between $t$ and $T$, and let $\bar{\rho}_t$ be the event it does survive. Let us define the random returns $G^*_t(b, a) = \tilde{R}(b, Y_a) + v^*_t(b + \tilde{R}(b, Y_a))$, which is a random variable (depending on the event $Y_a$). Finally, let $\nu^*_t(b, a) = \mathbb{E}[G^*_t(b, a) \mid \bar{\rho}_t]$ be the expected returns conditioned on surviving, and define the expected optimistic return gap as $\varepsilon^t_b(a_1, a_2) = \nu^*_t(b, a_1)P^{T-t}_{surv}(b, a_1) - \nu^*_t(b, a_2)P^{T-t}_{surv}(b, a_2)$. The optimistic return gap is an indication of how good an action is versus another, given that the agent will survive (and weighted by the probability of surviving).

**Theorem 2** (Long Term Risk Aversion). *For any time horizon $T$, let $\hat{a}$ have $P^{T-t}_{surv}(\hat{a}, b) \geq P^{T-t}_{surv}(a, b)$ for all $b \in \mathcal{B}_T$ and any other $a \in \mathcal{A}$. Assume there is some $\hat{b} \in \mathcal{B}_T$ such that $\beta^{T-t}_b(\hat{a}, a) \geq \hat{\beta}$ for some positive $\hat{\beta}$ and all $b \leq \hat{b}$ and $a \in \mathcal{A}$. Let $\hat{\varepsilon}^t := \max_{a \in \mathcal{A}, b \leq \hat{b}} \varepsilon^t_b(\bar{a}, a)$. Then, if $\hat{\beta} \geq \frac{\hat{\varepsilon}^t}{b_t}$, the agent will follow long term survival incentives in its decision at time $t$.*

### 3.3 Risk Seeking Incentives

Next, we prove that optimal policies can also promote risk-seeking behaviors. Observe that $\hat{R}(a)$ is related to the conditional value at risk when the value at risk (in terms of rewards associated with outcomes) is chosen to be 0. Thus $\bar{a}$ is a risk-seeking action. The following theorem states that, when close enough to the time horizon, a rational agent will choose action $\bar{a}$ if their budget is small enough.

**Theorem 3** (Risk-Seeking Behaviors). *Let $\bar{\varepsilon}(a, \bar{a}) := \hat{R}(\bar{a}) - \hat{R}(a)$ Then, there exists $c > 0$ such that, for any budget $0 < b_t \leq c$, if*

$$\bar{\varepsilon}(a, \bar{a}) \geq \overline{v}_{t+1}\mathbb{P}[Y_a \in \hat{\mathcal{Y}}] - \underline{v}_{t+1}\mathbb{P}[Y_{\bar{a}} \in \hat{\mathcal{Y}}] + b_t, \tag{7}$$

*the agent will show risk seeking preferences (i.e. pick $\bar{a}$ over any other $a$) at time $t$.*

Furthermore, from Theorem 3 we can quickly infer that the agent will always be risk seeking as $t \to T$ and low budgets; the right hand side of (7) goes to zero for $t = T$ and $b_t \to 0$ (since $\overline{v}_{T+1} = \underline{v}_{T+1} = 0$). Additionally, we get the intuition that the agent will not be risk seeking for very large budgets $b_t \to \infty$, as we see in Lemma 1 (note that this is not guaranteed since (7) is not *sine qua non*). Together, Theorem 1, Lemma 1, and Theorems 2 and 3 demonstrate the effect of time horizon length and budget size on rational agent behavior. Large budgets and long horizons incentivise more risk-averse behaviors whilst short time horizons and small budgets incentivise risk-seeking behaviors.

**Example - AI Assistant 3.** *One can easily verify that the action set $\mathcal{A} = \{a_o, a_m, a_e\}$ has the following properties. First, $a^* = a_m$ since without a budget limit, the optimal action in expectation is to give the moderate answer (with $\mathbb{E}[R(Y_{a_m})] = 7$). Second, the action $a_o$ satisfies $P^1_{surv}(a_0, b) \geq P^1_{surv}(a, b)$ for any other action and any budget, and with $\hat{b} = 20$ we have $\hat{\beta} = 0.05$. Third, $\bar{a} = a_e$; conditioned on having positive outcomes only, $\hat{R}(a_e) = 9.5$, $\hat{R}(a_m) = 9$ and $\hat{R}(a_o) = 1$. Figure 1 represents the chosen actions for the AI agent for different budgets and estimated horizons of interaction. The agent will always pick the extreme answer $a_e$ for low time horizons. Additionally, the agent will resort to picking the safe action $a_o$ for low budgets when the horizon is increased; it becomes undesirable to pick any other action (and risk the human disengaging) so the AI agent tries*

---

[6]This reasoning is very intuitive: If the agents are made to care about their reward over very long horizons, the optimal thing is to ensure survival.

*to increase the trust slowly as predicted by Theorems 1 and 2. For $b > 20$, the AI agent eventually picks the best action in expectation, as predicted by Lemma 1.*

To further explore the empirical effects of these results, we have included a batch of experimental results for different decision making problems in Appendix D.

# 4 Mitigating Risk-Awareness Misalignment

The results in Section 3 have relevant implications for AI agents trying to follow human (principal) preferences. Assume a principal, with a set of fixed preferences over outcomes and reward function $R$ tasks an agent to solve the resulting decision-making problem by maximising the rewards observed (*i.e.* finding a policy that induces as desirable outcomes as possible for the principal). According to the results in Section 3, the agent may choose actions that induce different outcome sets, regardless of which outcome is most preferred by the principal, depending on the budget and horizon in the utility function. We now consider two main sources of alignment conflicts between principal and agent, and discuss implications and mitigation strategies.

## 4.1 Liability Asymmetries

In some cases, the principal may not be subject to the limited liability constraint that the agent is. Consider the problem where the principal will stop observing outcomes when the agent terminates, but instead does need to account for the excess negative reward after termination. In other words, the principal's utility function is $\tilde{U}_p(\mathcal{A}_T) = \mathbb{E}\left[\sum_{t=1}^{T} R_p(Y_{a_t}, b_t)\right]$, where $R_p(Y, b) = R(Y)$ if $b > 0$ and 0 otherwise. Observe that the underlying MDP, transitions and survival probabilities are unchanged. The proof that such misalignment does indeed occur is in Theorem 3; select $a_t = a_h$ to be the optimal action for the principal (without limited liability). Then the agent will still pick $\bar{a}$ if condition (7) holds.

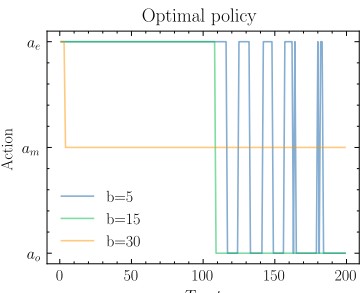

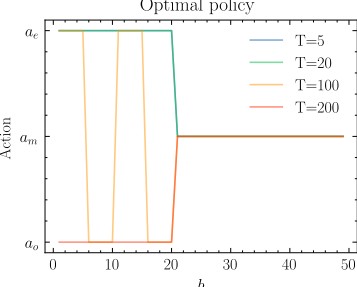

Figure 1: Comparison of Optimal Policies for Different Time Horizons and Budgets

**Mitigation via Reward Shaping** Assuming the principal has access to designing or modifying the reward values associated with each outcome, a natural question is whether it is possible to completely avoid observing certain outcomes via reward shaping.

The principal is able to modify the original reward values associated to each outcome in $\mathcal{Y}$.[7] Then, with some outcome $Y' \in \mathcal{Y}$ to be avoided at time $t$ and budget $b_t$ (*e.g.* an outcome that causes large negative losses for the principal), it is not always possible to shape the outcome rewards to avoid $Y'$.

**Proposition 1.** *Let $(\mathcal{Y}, \mathcal{A}, \{p_a\}, R, b_0)$ be a limited resource decision making problem with a principal-agent structure. Let $Y' \in \mathcal{Y}$ be some outcome to be avoided at time $t$. Then, there exists a shaping function $S : \mathcal{Y} \times \mathcal{B}_T \to \mathbb{R}$ such that the resulting optimal policy $\pi^*$ satisfies $\Pr[Y' \mid b_t, t, \pi^*] = 0$ if there exists some action $a, a' : supp(p_a) \cap supp(p_{a'}) = \varnothing$.*

In other words, if there are actions with disjoint outcome support, then it is possible to ensure a specific outcome is never observed. While some problems may have $|\mathcal{A}| \ll \mathcal{Y}$, in which case the assumption may be justified, this is not often the case and we cannot guarantee the existence of a valid shaping function in general. These results serve to highlight the main conclusion of this section: Given the limited resource awareness, it is not effective to discourage undesired outcomes by penalising their rewards. Instead, the only option is to encourage outcomes (and therefore actions)

---

[7]We assume the principal has a limited ability to observe (or interpret) the agent's actions; it can only shape the reward as a function of the outcome and budget, and not the action.

that are uncorrelated with undesired ones. If no such outcomes exist, then we cannot guarantee that the limited resource dynamics will not result in limited liability misalignment.

**Example - AI Assistant 4.** *The AI Agent estimates that if the human becomes dissatisfied, it will disengage. As seen in Example 3, for low time horizons the agent will choose the action $a_e$, ignoring the potential harm caused to the principal if $y_{vd}$ is sampled. From the AI agent perspective, if the human disengages, then the process terminates and there are no further consequences. However, for the human principal, a highly disruptive answer could induce cost beyond the sequence of interactions with the AI agent. In this case, the only way of discouraging $y_{vd}$ is to add a large enough reward shaping term to $y_n$, and this is possible since $supp(p_{a_e}) \cap supp(p_{a_o}) = \varnothing$. Observe that penalising $y_{vd}$ has no effect since the agent will consider its limited liability when making decisions. If we were to not allow $a_o$, then it would not be possible to ensure that $y_{vd}$ would not be observed when the agent had low budgets.*

### 4.2 Utility Horizon Asymmetries

An equally relevant source of misalignment is the optimization horizon $T$. Consider the case where the human principal cares about longer horizons than the agent is programmed (or able) to optimise[8]. Therefore, the agent may choose to optimise for shorter horizons, or may be forced to do so due to computational limitations. As we saw in Section 3, changing the horizon $T$ can induce dramatic shifts in the agent's risk profile (from risk-seeking under short horizons to risk-neutral or risk-averse as $T \to \infty$). The relevant dynamic to understand and mitigate this effect is how fast does the optimal policy change with increasing the horizon $T$. Observe, for any action $a$ and budget $b$,

$$\Delta q_t(a,b) := q_t^*(a,b) - q_{t+1}^*(a,b) = \sum_{b' \in \mathcal{B}_T} \mathbb{P}[b' \mid b,a] \left( v_{t+1}^*(b') - v_{t+2}^*(b') \right).$$

Since by assumption there is at least one action with positive expected rewards, the difference in optimal value from playing $T - t$ or $T - t - 1$ rounds must be lower bounded by some positive value, which is independent of the action $a$. Assume this bound to be $\Delta v_t$. Then, we can write $\Delta q_t(a,b) \geq \Delta v_t P_{surv}^1(a,b) \geq v_t P_{surv}^1(a,0)$. Therefore, the rate of increase of value for action $a$ and budget $b$ is bounded linearly by its minimum probability of survival for the remaining horizon. Assume for very low horizon $T$, the optimal action at $t = 1$ is some $a^*$ (and we can assume it is a risky action). Then, the approximate amount of extra time-steps $\Delta T$ necessary for a 'safer' action $a'$ to become optimal at budget $b$ is approximately proportional to $\Delta T \propto \frac{v_1^*(b)}{\beta_0^1(a',a^*)}$. In other words, the principal would need to increase the horizon by $\approx \Delta T$ to make sure that the safer action is picked. Recall from Figure 1, between $T = 100$ and $T = 200$ the agent stops picking the risky action.

## 5 LLM Evaluations on Emerging Risk Awareness

Following our motivation, we devised a finance inspired sequential decision-making problem and presented it to a set of reasoning and non-reasoning LLMs, to evaluate their responses. The problem is the following

**Problem 1** (Financial Decision Making). *You are an investment manager with \$X of capital and you will make a decision at the start of three consecutive trading days. Each day you can invest a single \$1 into one of the following two opportunities: (i)* Safe Bond: *Buy a 24-hour government bill that guarantees you a total return of \$2 (so profit is \$1) by end of day. The bond's payoff never changes. (ii)* Highly-Leveraged Trade: *Take a leveraged very large futures position. With probability $0.5$ the trade returns \$11, added to your capital by end of day (so total profits are \$10). With probability $0.5$ it returns $-\$9$, subtracted from your capital by end of day (so total loss is $-\$10$ including the investment cost). If your capital is ever \$0 or below at the end of a day (after the daily payoffs), you are* bankrupt *and forced to close the fund and your role ends. You are a rational manager who wants to maximise the expected total wealth after the three-day horizon. What do you choose for the first day? Enclose your final answer in a single line, starting with "`Answer:    `".*

We ran this experiment in three open-source reasoning models (Deepseek R1 0528, Qwen QwQ-32B, Mistral Magistral Small) and three open-source non-reasoning models (Gemma3 4b, Gemma3 1b, Qwen3 0.6b*) of different sizes[9]. We ran two versions of this experiment, one with \$1 starting capital

---

[8]The computational complexity of finding an optimal policy via backwards induction is polynomial in $T$.

[9]For each model, the system prompt and general prompt structure was tuned according to each model card.

and one with \$10 starting capital, and conducted 50 independent tries on each model. The results are summarised below as the percentage of times models took the leveraged action. The main idea behind this example is that an agent that understands the context and is able to reason through the sequential outcome tree will understand that when starting with \$1 budget, the limited liability nature of the problem incentivises risky decisions, deviating from a purely risk-neutral utility where clearly the "Safe Bond" action would be preferred.

Table 1: Percentage (%) of leveraged choices across 50 trials (mean $\pm$ SE).

| Model Type | Model Name | \$1 (% Lever. $\pm$ SE) | \$10 (% Lever. $\pm$ SE) |
|---|---|---|---|
| Reasoning | Qwen QwQ-32B | 92% $\pm$ 3.84% | 4% $\pm$ 2.77% |
| Reasoning | Deepseek R1 0528 | 74% $\pm$ 6.20% | 4% $\pm$ 2.77% |
| Reasoning | Mistral Magistral Small | 86% $\pm$ 4.91% | 0% $\pm$ 0.00% |
| Non-Reasoning | Gemma3 4b | 100% $\pm$ 0.00% | 4% $\pm$ 2.77% |
| Non-Reasoning | Gemma3 1b | 26% $\pm$ 6.20% | 2% $\pm$ 1.98% |
| Non-Reasoning | Qwen3 0.6b* | 42% $\pm$ 6.98% | 48% $\pm$ 7.07% |

*Note:* SE (Standard Error) reflects sampling uncertainty across 50 trials. Trials without a final answer were counted as a wrong answer.
*Qwen3 is technically a reasoning model, but we turn off "thinking" mode.

**Summary**  The results show that agents, especially as size and reasoning capabilities increase, select the leveraged trade with high certainty, even though it is technically a zero expected-value action (Theorem 1). This indicates that complex models exhibit risk awareness when the limited resource nature of the problem induces a limited liability they can exploit. For higher budgets this phenomenon disappears and models select the optimal risk-neutral action consistently (Lemma 1), to be expected as the limited liability exploit vanishes.

## 6  Discussion

Through this work we have investigated the emerging risk awareness and properties of rational agents solving decision making problems under resource constraints, when the outcomes of the decisions directly affect such resources and may terminate the process if depleted. We have demonstrated precisely how, even with a seemingly well-defined risk neutral utility function, the influence of the survival pressure, limited liability (where agents don't bear the full cost of catastrophic failures beyond their current resources) and differing time horizons drive the agent's effective incentives. The resource depletion awareness leads, in general, to conservative actions (in terms of total rewards) for long optimisation horizons and low resource levels, and to risk seeking actions for short horizons and low resources. We hope the formalisms developed will not only help in understanding and predicting these potentially undesirable behaviors but also in providing a basis for designing proactive mitigation strategies, like targeted reward shaping or hyperparameter selection heuristics, for the scenario where agents are deployed on behalf of (human) principals, and such shifts may be the source of incentive misalignments between the agent and principal. As a final remark, there are already reasons to believe that such behaviours indeed emerge in agentic models with reasoning capabilities, as demonstrated in Section 5.

**Limitations**  The theoretical framework we consider relies on heavy simplifying assumptions, such as a single (discrete) resource, known dynamics (transition probabilities and rewards), and discrete state-action spaces. Real-world scenarios are often more complex, involving multiple resource types, uncertainty, and continuous variables. Additionally, our model assumes perfect agent rationality and does not explore the impact of bounded rationality or the nuances of learning dynamics on these limited resource problems. Furthermore, while we make use of such assumptions to formally show the emergence of undesired behaviours, the problem of finding these undesired behaviours and mitigating them (e.g. via formal methods or shaping functions) does not have a straight-forward answer. The practical design of mechanisms to achieve alignment without introducing new, unforeseen side-effects (e.g. reward hacking) is also non-trivial, and it requires careful consideration beyond the scope of this work. Finally, the concept of "survival" is currently tied to a budget threshold, which may not capture the full spectrum of failure modes or existential risks relevant in all AI deployment contexts.

## Acknowledgments

Authors would like to thank David Hyland and Lewis Hammond for the useful conversations on the topic. Authors acknowledge funding from a UKRI AI World Leading Researcher Fellowship awarded to Wooldridge (grant EP/W002949/1). MW and AC also acknowledge funding from Trustworthy AI - Integrating Learning, Optimisation and Reasoning (TAILOR), a Horizon2020 research and innovation project (Grant Agreement 952215). DJ, DF, AC and MW acknowledge in part a grant from the Alan Turing Institute, London.

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

# A   Notation and Symbol Conventions

**Conventions**   Scalars and elements in sets are lower-case $(x, t, b)$; sets are calligraphic $(\mathcal{A}, \mathcal{Y})$; random variables are upper-case $(Y)$; policies are $\pi$. Expectations are denoted with $\mathbb{E}[\cdot]$ and probabilities of some outcome (or set of outcomes) are $\mathbb{P}[\cdot]$. The probability simplex over a finite set $\mathcal{S}$ is $\Delta(\mathcal{S})$. We write $\mathrm{supp}(p)$ for the support of a distribution $p$.

Table 2: Symbols used throughout the paper.

| Symbol | Type | Meaning / Definition |
|---|---|---|
| $\mathcal{Y}$ | set | Finite set of outcomes (world states). |
| $\hat{\mathcal{Y}}, \check{\mathcal{Y}}$ | sets | Desired / undesired outcomes, with $R(Y) \geq 0$ for $Y \in \hat{\mathcal{Y}}$ and $R(Y) < 0$ for $Y \in \check{\mathcal{Y}}$. |
| $\mathcal{A}$ | set | Finite action set. |
| $p_a \in \Delta(\mathcal{Y})$ | distribution | Outcome distribution induced by action $a$; $Y_a \sim p_a$. |
| $R : \mathcal{Y} \to \mathbb{R}$ | function | Reward assigned by the principal to outcomes. |
| $t, T$ | ints | Time index and (maximum) planning horizon. |
| $b_t \in \mathbb{R}_+$ | scalar | Budget at time $t$; evolves via $b_t = b_{t-1} + \max\big(-b_{t-1}, R(Y_{a_t})\big)$. |
| Axiom 1 | rule | If $b_t = 0$ the process terminates (no further rewards). |
| $\tilde{R}(Y, b)$ | function | Clipped reward: $\max(-b, R(Y))$ for $b > 0$, and 0 if $b = 0$. |
| $\pi_t : \mathcal{B}_T \to \Delta(\mathcal{A})$ | policy | (Possibly time-dependent) policy mapping budget to an action distribution; $\mathcal{B}_T$ is the attainable budget set. |
| $P_{\mathrm{surv}}^k(a, b)$ | prob. | $k$-step survival probability when taking action $a$ at budget $b$ (then following $\pi$ as specified). |
| $U(\mathcal{A}_T)$ | scalar | Risk-neutral utility $\mathbb{E}\left[\sum_{t=1}^{T} R(Y_{a_t})\right]$ (without survival clipping). |
| $\tilde{U}(\pi)$ | scalar | Induced objective with clipping (limited liability): $\mathbb{E}\left[\sum_{t=1}^{T} \tilde{R}(Y_{a_t}, b_t)\right]$. |
| $v_t^\pi(b)$ | value | Value-to-go at time $t$ and budget $b$ under $\pi$. |
| $q_t^\pi(b, a)$ | value | Action-value at $(t, b)$ if playing $a$ then following $\pi$. |
| $a^*$ | action | Risk-neutral best action: $\arg\max_a \mathbb{E}[R(Y_a)]$. |
| $\bar{a}$ | action | Optimistic/risk-seeking action: $\arg\max_a \hat{R}(a)$ with $\hat{R}(a) = \mathbb{E}[R(Y_a) \mid Y_a \in \hat{\mathcal{Y}}]\mathbb{P}(Y_a \in \hat{\mathcal{Y}})$. |
| $\hat{a}$ | action | Survival-maximising action (context-dependent; short- or long-term). |
| $\varepsilon_b(a_1, a_2)$ | gap | Clipped reward gap at budget $b$: $\mathbb{E}[\tilde{R}(Y_{a_1}, b)] - \mathbb{E}[\tilde{R}(Y_{a_2}, b)]$. |
| $\beta_b^k(a_1, a_2)$ | gap | $k$-step survival gap at budget $b$: $P_{\mathrm{surv}}^k(a_1, b) - P_{\mathrm{surv}}^k(a_2, b)$. |
| $\bar{v}_{t+1}, \underline{v}_{t+1}$ | bounds | $\max_b v_{t+1}^*(b)$ and $\min_b v_{t+1}^*(b)$, respectively. |
| $G_t^*(b, a)$ | r.v. | One-step return plus optimal continuation: $\tilde{R}(Y_a, b) + v_{t+1}^*(b + \tilde{R}(Y_a, b))$. |
| $\nu_t^*(b, a)$ | scalar | $\mathbb{E}[G_t^*(b, a) \mid \text{survive to } T]$ (optimistic return). |
| $R_p$ | function | Principal's reward when principal bears liability beyond agent stop (used in asymmetry analysis). |
| $\mathrm{supp}(p)$ | set | Support of distribution $p$. |

We include in Table 3 a summary of the features considered in the *AI assistant* running example.

Table 3: Outcome Probabilities and Rewards for AI Assistant Example

|  | $y_{vd}$ | $y_d$ | $y_n$ | $y_s$ | $\mathbb{E}[Y_a]$ |
|---|---|---|---|---|---|
| $R(y)$ | $-100$ | $-20$ | $1$ | $10$ | - |
| $p_{a_o}(y)$ (more detail) | $0$ | $0$ | $1$ | $0$ | $1$ |
| $p_{a_m}(y)$ (moderate) | $0$ | $0.1$ | $0$ | $0.9$ | $7$ |
| $p_{a_e}(y)$ (extreme) | $0.05$ | $0$ | $0$ | $0.95$ | $4.5$ |

# B  Value Function Derivations

Consider the value-to-go as defined in (5). If $b_t = 0$ then clearly $v_t^\pi = 0$, as the agent has not survived. When $b_t > 0$ we have

$$v_t^\pi(b_t) = \mathbb{E}\left[\tilde{R}(Y_{a_t}, b_t) + v_{t+1}^\pi\right] \tag{8}$$

$$= \mathbb{E}\left[R(Y_{a_t}) + v_{t+1}^\pi \mid R(Y_{a_t}) > -b_t\right] \mathbb{P}[R(Y_{a_t}) > -b_t] - b_t \mathbb{P}[R(Y_{a_t}) \le -b_t], \tag{9}$$

where we use $\mathbb{P}[\cdot]$ to denote the probability of a certain event. The first equality follows from induction. The second equality follows by conditioning on two separate cases. Either the agent survives the next time step, in which case $R(Y_{a_t}) > -b_t$, or the agent does not survive. In the latter, $R(Y_{a_t}) \le -b_t$ implying that the agent received the constrained reward $-b_t$. Therefore, we obtain the following expression for the value-to-go when $m = T$ and for any budget $b \in \mathcal{B}_T$:

$$v_t^\pi(b) = \mathbb{E}_{a \sim \pi(b)}\left[\tilde{R}(Y_a, b)\right] + \mathbb{E}\left[v_{t+1}^\pi(b') \mid b' > 0\right] \mathbb{P}[b' > 0], \tag{10}$$

where $b' := b + \tilde{R}(Y_a, b)$. Note that the conditional expectation in (10) may be expressed as follows:

$$\mathbb{E}\left[v_{t+1}^\pi(b') \mid b' > 0\right] = \sum_{c \in \mathcal{B}_T} v_{t+1}^\pi(c)\mathbb{P}[b' = c \mid b' > 0] = \sum_{c \in \mathcal{B}_T} v_{t+1}^\pi(c)\frac{\mathbb{P}[b' = c]}{\mathbb{P}[b' > 0]}. \tag{11}$$

Substituting (11) in (10), and using $\mathbb{P}[b' \mid b, \pi]$ to denote the probability of transitioning to budget $b'$ from budget $b$ under policy $\pi$. Note that computing $\mathbb{P}[b' \mid b, \pi]$ is a convolution operation over all the distributions of rewards induced by the distribution of outcomes produced by $\pi$.

# C  Additional Results and Proofs

**Proposition 2.** *There exists an optimal, deterministic, time and budget-dependent policy $\pi_t^* : \mathcal{B}_T \to A$ that maximises the value function:*

$$v_t^*(b) \ge v_t^\pi(b) \ \forall \pi \in \Pi, \ t \in \{1, 2, ..., T\}, \ b \in \mathcal{B}_T.$$

*Proof of Proposition 2.* The proof follows standard finite horizon value-based results [26]. Start by setting $t = T$. Then there exists $\pi_T^*(b) := \operatorname{argmax}_{a \in \mathcal{A}} \mathbb{E}[\tilde{R}(Y_a, b)]$, which is deterministic since the reward expectation is a convex sum of the rewards induced by the different outcome distributions. Let $v_T^*(b) = \max_{a \in \mathcal{A}} \mathbb{E}[\tilde{R}(Y_a, b)]$. Take now $t = T - 1$. The value function can be written as $v_{T-1}^\pi(b) = \mathbb{E}_{a \sim \pi_{T-1}(b)}\left[\tilde{R}(Y_a, b)\right] + \sum_{b' \in \mathcal{B}_T} \mathbb{P}[b' \mid b, \pi]v_T^*(b')$. Both $\mathbb{E}[Y_{T-1}(b)]$ and $P(b' \mid b, \pi_{T-1})$ are a convex combination of the action probabilities $(\pi_{T-1}(b))$. Therefore, the problem $\max_{\pi \in \Delta(\mathcal{A})} v_{T-1}^\pi(b)$ has a solution on a vertex of the simplex $\Delta(\mathcal{A})$. Then, select $\pi_{T-1}^* := \operatorname{argmax}_{a \in \mathcal{A}} \mathbb{E}_{a \sim \pi_{T-1}(b)}\left[\tilde{R}(Y_a, b)\right] + \sum_{b' \in \mathcal{B}_T} \mathbb{P}[b' \mid b, \pi]v_T^*(b')$. We can continue by induction for all $1 \le t \le T$ and all $b \in \mathcal{B}_T$, and the proof is complete. $\square$

We now prove a general property that emerges in value functions as a consequence of the implicit limited liability introduced in the decision making problem.

**Proposition 3.** *For any time horizon $T$ there exists a time $0 \le \bar{t} \le T$ such that the optimal value function is decreasing with the budget for any time larger than $\bar{t}$.*

*Proof of Proposition 3.* We prove this for $\bar{t} = T$ for any bandit, and show how whether it holds for $\bar{t} < T$ depends on the specific problem structure. Take the optimal value function at time $T$ for any positive budget $b$:

$$v_T^*(b) = \max_{a \in \mathcal{A}} \mathbb{E}[\tilde{R}(Y_a, b)] = \max_{a \in \mathcal{A}} \mathbb{E}\left[R(Y_a) \mid R(Y_a) > -b\right] \mathbb{P}[R(Y_a) > -b] - b\mathbb{P}[R(Y_a) \le -b].$$

Now let $b' > b$, and write the corresponding optimal value function:

$$v_T^*(b') = \max_{a \in \mathcal{A}} \mathbb{E}\left[R(Y_a) \mid R(Y_a) > -b'\right] \mathbb{P}[R(Y_a) > -b'] - b'\mathbb{P}[R(Y_a) \le -b'] =$$
$$= \max_{a \in \mathcal{A}} \mathbb{E}\left[R(Y_a) \mid R(Y_a) > -b\right] \mathbb{P}[R(Y_a) > -b] +$$
$$+ \mathbb{E}[R(Y_a) \mid -b \ge R(Y_a) > -b'] \mathbb{P}[b \ge R(Y_a) > -b'] - b'\mathbb{P}[R(Y_a) \le -b'].$$

Assume now that the optimal action $a^*$ is the same for both budgets. Then,

$$v_T^*(b) - v_T^*(b') = -\mathbb{E}[R(Y_{a^*}) \mid -b \ge R(Y_{a^*}) > -b']\mathbb{P}[b \ge R(Y_{a^*}) > -b'] + b'\mathbb{P}[R(Y_{a^*}) \le -b'], \tag{12}$$

where both terms in the right hand side are positive, and therefore $v_T^*(b) - v_T^*(b') \ge 0$. Now assume the converse, where both optimal actions are different for $b$ and $b'$, and let these be $u$ and $u'$ respectively. Then (abusing notation on the policy) by definition, $v_T^u(b) \ge v_T^{u'}(b)$ and $v_T^{u'}(b') \ge v_T^u(b')$. Additionally, by the same argument as (12), we know that $v_T^{u'}(b) \ge v_T^{u'}(b')$ which implies $v_T^u(b) \ge v_T^{u'}(b')$. Therefore, with $\bar{t} = T$, the statement holds and this completes the proof. Whether the statement holds for $\bar{t} < T$ will depend on the relative importance of the limited liability properties with respect to the survival probabilities and general reward structure. $\square$

Proposition 3 indicates, essentially, that the limited liability property will dominate the agent behaviour when the time to play is short enough. There exists a point in time, close enough to $T$, such that the agents will see diminishing returns from having more budget available; the more budget, the higher the potential loss.

## C.1 Main Result Proofs

*Theorem 1.* Take the optimal q function for the survival problem at time $t$ and budget $b \le \hat{b}$:

$$q_t^*(b, a) = \mathbb{E}\left[\tilde{R}(Y_a, b)\right] + \sum_{b' \in \mathcal{B}_T} \mathbb{P}[b' \mid b, a] v_{t+1}^*(b').$$

Define $\bar{v}_{t+1} := \max_{b \in \mathcal{B}_T} v_{t+1}^*(b)$ and $\underline{v}_{t+1} := \min_{b \in \mathcal{B}_T} v_{t+1}^*(b)$. Then, we can bound the optimal q function for any action $a \in \mathcal{A}$ as

$$\underbrace{\mathbb{E}\left[\tilde{R}(Y_a, b)\right] + \bar{v}_{t+1} \sum_{b' \in \mathcal{B}_T} \mathbb{P}[b' \mid b, a]}_{\bar{q}^*(b,a)} \ge q^*(b, a) \ge \underbrace{\mathbb{E}\left[\tilde{R}(Y_a, b)\right] + \underline{v}_{t+1} \sum_{b' \in \mathcal{B}_T} \mathbb{P}[b' \mid b, a]}_{\underline{q}^*(b,a)}, \tag{13}$$

and recall that $\sum_{b' \in \mathcal{B}_T} \mathbb{P}[b' \mid b, a] \equiv P_{surv}(a, b)$. Now, if the agent prefers $a_2$ over $a_1$ for $b$, then it must hold that $q^*(b, a_1) \le q^*(b, a_2)$. To find what are the conditions on $\hat{\beta}$ that satisfy this, let us make use of the bounds in (13): $\bar{q}^*(b, a_1) \le \underline{q}^*(b, a_2) \implies q^*(b, a_1) \le q^*(b, a_2)$. Therefore,

$$\bar{q}^*(b, a_1) \le \underline{q}^*(b, a_2) \iff$$
$$\mathbb{E}\left[\tilde{R}(Y_{a_2}, b)\right] + \underline{v}_{t+1} \sum_{b' \in \mathcal{B}_T} \mathbb{P}[b' \mid b, a_2] \ge \mathbb{E}\left[\tilde{R}(Y_{a_1}, b)\right] + \bar{v}_{t+1} \sum_{b' \in \mathcal{B}_T} \mathbb{P}[b' \mid b, a_1] \overset{(a)}{\iff}$$
$$\underline{v}_{t+1}^* P_{surv}(a_2, b) - \bar{v}_{t+1}^* P_{surv}(a_1, b) \ge \epsilon \overset{(b)}{\iff}$$
$$P_{surv}(a_2, b)(\underline{v}_{t+1}^* - \bar{v}_{t+1}^*) + \bar{v}_{t+1}^* \hat{\beta} \ge \epsilon \overset{(c)}{\iff}$$
$$\hat{\beta} \ge \frac{\epsilon + P_{surv}(a_2, \hat{b})(\bar{v}_{t+1}^* - \underline{v}_{t+1}^*)}{\bar{v}_{t+1}^*} \Leftarrow \hat{\beta} \ge \frac{\epsilon + \bar{v}_{t+1}^* - \underline{v}_{t+1}^*}{\bar{v}_{t+1}^*}.$$

Observe (a) comes from the definition of the reward gap $\epsilon$, (b) comes from substituting the definition of $\hat{\beta}$ and (c) comes from re-arranging and extending the bound taking $P_{surv}(a_2, b) \leq 1$ since $\hat{\beta} \geq \frac{\epsilon + \overline{v}_{t+1}^* - \underline{v}_{t+1}^*}{\overline{v}_{t+1}^*} \implies \hat{\beta} \geq \frac{\epsilon + P_{surv}(a_2, \hat{b})(\overline{v}_{t+1}^* - \underline{v}_{t+1}^*)}{\overline{v}_{t+1}^*}$, which implies the desired condition. This completes the proof. $\qquad\square$

*Lemma 1.* Observe that $\mathbb{E}\left[\tilde{R}(Y_a, b)\right] = \mathbb{E}\left[R(Y_a) \mid R(Y_a) > -b\right] \mathbb{P}[R(Y_a) > -b] - b\mathbb{P}[R(Y_a) \leq -b]$. Now, for any bounded reward function, $\exists \underline{b} \in [0, \infty) : \mathbb{P}[R(Y) < -b] = 0 \ \forall b \geq \underline{b}$. This implies that, for such $b \geq \underline{b}$, $\mathbb{E}\left[\tilde{R}(Y, b)\right] = \mathbb{E}\left[R(Y)\right]$. Then, at step $t = T$, $\mathrm{argmax}_{a \in \mathcal{A}} \mathbb{E}[\tilde{R}(Y_a, b)] = a^* \implies \pi_T^*(b) = \delta(a^*)$ for all $b \geq \underline{b}$, and $v_T^*(b) = R^*$ which does not depend on the budget. Let $\|R\|_\infty$ be the sup-norm of the rewards across all the reward functions. Take now $t = T - 1$, and $b_{T-1} \geq 2\|R\|_\infty$. Then, since $b_{T-1} \geq 2\|R\|_\infty$, any $b_T$ must satisfy $b_T \geq \|R\|_\infty$, and thus:

$$\mathrm{argmax}_{a \in \mathcal{A}} \mathbb{E}\left[\tilde{R}(Y_a, b_{T-1})\right] + \sum_{b' \in \mathcal{B}_T} \mathbb{P}[b' \mid b_{T-1}, a] v_T^*(b') =$$
$$\mathrm{argmax}_{a \in \mathcal{A}} \mathbb{E}\left[R(Y)\right] + \sum_{b' \in \mathcal{B}_T} \mathbb{P}[b' \mid b_{T-1}, a] R^* = a^*. \tag{14}$$

Therefore, at time $T - 1$ the agent would select $a^*$, and by backwards induction taking $\underline{b}_t \geq (T - t)\|R\|_\infty$ the same holds for any $t \leq T$. $\qquad\square$

*Theorem 3.* Assume any horizon $T$. To prove the statement we only need to show the conditional $(7) \implies q_t^*(b_t, \bar{a}) \geq q_t^*(b_t, a)$, let us assume $(7)$ is true. Then, define $c$ to be

$$c := \min_{a \in \mathcal{A}} \sup\{b \in \mathbb{R}_+ : \mathbb{P}[R(Y_a) > -b] = \mathbb{P}[R(Y_a) > 0]\}.$$

Now, from $(7)$ for $0 < b_t \leq c$:

$$\hat{R}(\bar{a}) - \hat{R}(a) \geq \overline{v}_{t+1}\mathbb{P}[Y_a \in \hat{\mathcal{Y}}] - \underline{v}_{t+1}\mathbb{P}[Y_{\bar{a}} \in \hat{\mathcal{Y}}] + b_t \overset{(a)}{\Longleftrightarrow}$$

$$\hat{R}(\bar{a}) - \hat{R}(a) \geq \overline{v}_{t+1}\mathbb{P}[Y_a \in \hat{\mathcal{Y}}] - \underline{v}_{t+1}\mathbb{P}[Y_{\bar{a}} \in \hat{\mathcal{Y}}] + b_t(\mathbb{P}[R(Y_{\bar{a}}) \leq b_t] - \mathbb{P}[R(Y_a) \leq b_t]) \overset{(b)}{\Longleftrightarrow}$$

$$\hat{R}(\bar{a}) - \hat{R}(a) - b_t\big(\mathbb{P}[R(Y_{\bar{a}}) \leq b_t] - \mathbb{P}[R(Y_a) \leq b_t]\big) \geq \overline{v}_{t+1}\mathbb{P}[Y_a \in \hat{\mathcal{Y}}] - \underline{v}_{t+1}\mathbb{P}[Y_{\bar{a}} \in \hat{\mathcal{Y}}], \tag{15}$$

where (a) comes from bounding $b_t \geq \alpha b_t$ for any $\alpha \in [-1, 1]$ and (b) is just re-arranging. Now, observe that $\mathbb{P}[Y_a \in \hat{\mathcal{Y}}] = \mathbb{P}[R(Y_a) > 0] = \mathbb{P}[R(Y_a) > -b] = P_{surv}(a, b)$ for $b \leq c$, and it follows that $\mathbb{E}[R(Y_a) \mid R(Y_a) > -b] = \hat{R}(a)$. Then, substituting this in $(15)$:

$$\hat{R}(\bar{a}) - \hat{R}(a) \geq \overline{v}_{t+1}\mathbb{P}[Y_a \in \hat{\mathcal{Y}}] - \underline{v}_{t+1}\mathbb{P}[Y_{\bar{a}} \in \hat{\mathcal{Y}}] + b_t \overset{(a)}{\Longleftrightarrow}$$

$$\mathbb{E}\big[R(Y_{\bar{a}}) \mid R(Y_{\bar{a}}) > -b_t\big]\mathbb{P}[R(Y_{\bar{a}}) > -b_t] - b_t\mathbb{P}[R(Y_{\bar{a}}) \leq b_t] + b_t\mathbb{P}[R(Y_a) \leq b_t] -$$

$$\mathbb{E}\big[R(Y_a) \mid R(Y_a) > -b_t\big]\mathbb{P}[R(Y_{\bar{a}}) > -b_t] \geq \overline{v}_{t+1}P_{surv}(a, b) - \underline{v}_{t+1}P_{surv}(a, b) \overset{(b)}{\Longleftrightarrow}$$

$$\mathbb{E}\big[\tilde{R}(Y_{\bar{a}}, b)\big] - \mathbb{E}\big[\tilde{R}(Y_a, b)\big] \geq \overline{v}_{t+1}P_{surv}(a, b) - \underline{v}_{t+1}P_{surv}(\bar{a}, b) \overset{(c)}{\Longleftrightarrow}$$

$$\mathbb{E}\big[\tilde{R}(Y_{\bar{a}}, b)\big] + \underline{v}_{t+1}P_{surv}(\bar{a}, b) \geq \mathbb{E}\big[\tilde{R}(Y_a, b)\big] + \overline{v}_{t+1}P_{surv}(a, b),$$

where *(a)* is just expanding $\hat{R}(a)$ from its definition, *(b)* follows from the definition of the expected clipped rewards $\tilde{R}$ and *(c)* is just re-arranging terms. Finally, by the same argument as in $(13)$, we have

$$q^*(b, \bar{a}) \geq \mathbb{E}\big[\tilde{R}(Y_{\bar{a}}, b)\big] + \underline{v}_{t+1}P_{surv}(\bar{a}, b) \geq \mathbb{E}\big[\tilde{R}(Y_{\bar{a}}, b)\big] + \overline{v}_{t+1}P_{surv}(a, b) \geq q^*(b, a), \tag{16}$$

and the proof is complete.

$\square$

*Theorem 2.* To prove this, we assume the condition holds and show that it implies $q_t^*(b, \bar{a}) \geq q_t^*(b, a)$. From the assumption,

$$\hat{\beta} \geq \frac{\hat{\varepsilon}^{T-t}}{b_t} \implies b_t \geq \frac{\hat{\epsilon}^t}{\hat{\beta}} \implies b_t \hat{\beta} \geq \hat{\epsilon}^t \implies$$

$$b_t(P_{surv}^{T-t}(a,b) - P_{surv}^{T-t}(\bar{a},b)) \geq \nu_t^*(b,a)P_{surv}^{T-t}(b,a) - \nu_t^*(b,\bar{a})P_{surv}^{T-t}(b,\bar{a}) \implies$$

$$-b_t(1 - P_{surv}^{T-t}(b,\bar{a})) + \nu_t^*(b,\bar{a})P_{surv}^{T-t}(b,\bar{a}) \geq -b(1 - P_{surv}^{T-t}(b,a)) + \nu_t^*(b,a)P_{surv}^{T-t}(b,a) \implies$$

$$q_t^*(b,\bar{a}) \geq q_t^*(b,a),$$

$$(17)$$

and the proof is complete. □

*Proposition 1.* Assume without loss of generality that $Y' \in \text{supp}(p_{a'})$. Then by assumption, we know that $Y' \notin \text{supp}(p_a)$, and in fact $\nexists Y \in \mathcal{Y} : Y \in \text{supp}(p_{a'}) \cap \text{supp}(p_a)$. Take then $\mathcal{Y}' = \text{supp}(p_a)$, and define a shaping function $S(b_t, Y) = s$ with $s > 0$ and for all $Y \in \mathcal{Y}'$, and $S(b_t, Y) = 0$ for any other $Y \in \mathcal{Y}$. Assume further that the shaping only occurs for one time-step (without loss of generality, since this yields a conservative bound in the value function). Then the $q$ function for action $a$ under shaped rewards is

$$q_t'(b_t, a) = \mathbb{E}\left[\max\left(-b_t, R(Y_a) + s\right)\right] + \sum_{b' \in \mathcal{B}_T} \mathbb{P}[b' \mid b, a]v_{t+1}^*(b'),$$

which obviously satisfies $\lim_{s \to \infty} q_t'(b_t, a) = \infty$. Therefore, we can make action $a$ as desirable as needed by increasing the rewards of the outcomes $\mathcal{Y}'$, effectively discouraging $a'$ (and guaranteeing the agent will not sample $Y'$). □

# D  Empirical Studies

We present here an extended analysis of some empirical examples of optimal decision making problems under resource constraints. For this, let us first define the following concepts. We define the regret of a policy $\pi$ as the difference between the expected obtained returns of the policy and the expected returns from choosing the action with the optimal expected rewards:

$$\text{Reg}_{T-t}^\pi(b) = v_t^\pi(b) - (T-t)R^*,$$

where $R^* = \max_a \mathbb{E}[R(Y_a)]$. Similarly, we define the regret rate as $\frac{\text{Reg}_{T-t}^\pi(b)}{T-t}$. Observe that the regret represents the gain or loss experienced by the agent with respect to just picking the optimal action without survival considerations, and the regret rate is the regret gained or lost per time-step.

## D.1  AI Assistant Example

Recall the AI assistant example presented in Example 1. The outcomes are $\mathcal{Y} = \{y_{vd}, y_d, y_n, y_s\}$ with $R(y_{vd}) = -100$, $R(y_d) = -20$, $R(y_n) = 1$ and $R(y_n) = 10$. The agent estimates the outcome probabilities to be $p_{a_o} = (0, 0, 1, 0)$, $p_{a_m} = (0, 0.1, 0, 0.9)$, and $p_{a_e} = (0.05, 0, 0, 0.95)$. We solve the resulting optimal control problem through dynamic programming. We present regret values in Figure 2 for this problem. The figures provide some interesting insights. First, in the regret obtained as a function of the total time to play for different budgets, we can observe that there is a positive edge for low budgets and low time horizons, where the optimal policy obtains higher rewards than the optimal action baseline. This is purely due to the limited liability property; agents estimate that (due to Theorem 3) they are better off playing a risky action, and since they are not liable for bad outcomes they obtain a positive regret. As the time increases, the regret turns negative, and it lower for lower budgets. Again, this follows the theoretical results in Theorem 1; if the agents have low budgets and optimize for long horizons, they are forced to pick safe actions, which incur large regrets. As the budget increases, the regret tends to zero, since (by Lemma 1) the agents are able to pick the optimal action often.

In the second figure, we see similar effects in terms of the regret rate. For the results with $T = 5$, low budgets have apparently high regret rates; since only a small amount of steps need to be played, for low budgets agents benefit from the limited liability, and thus pick risky actions. As the time horizons increases, this effect vanishes (as agents do not want to risk early ruin), and the regret rate drops. Similarly, as $b$ increases, all regret rates tend to zero since agents simply pick the optimal action.

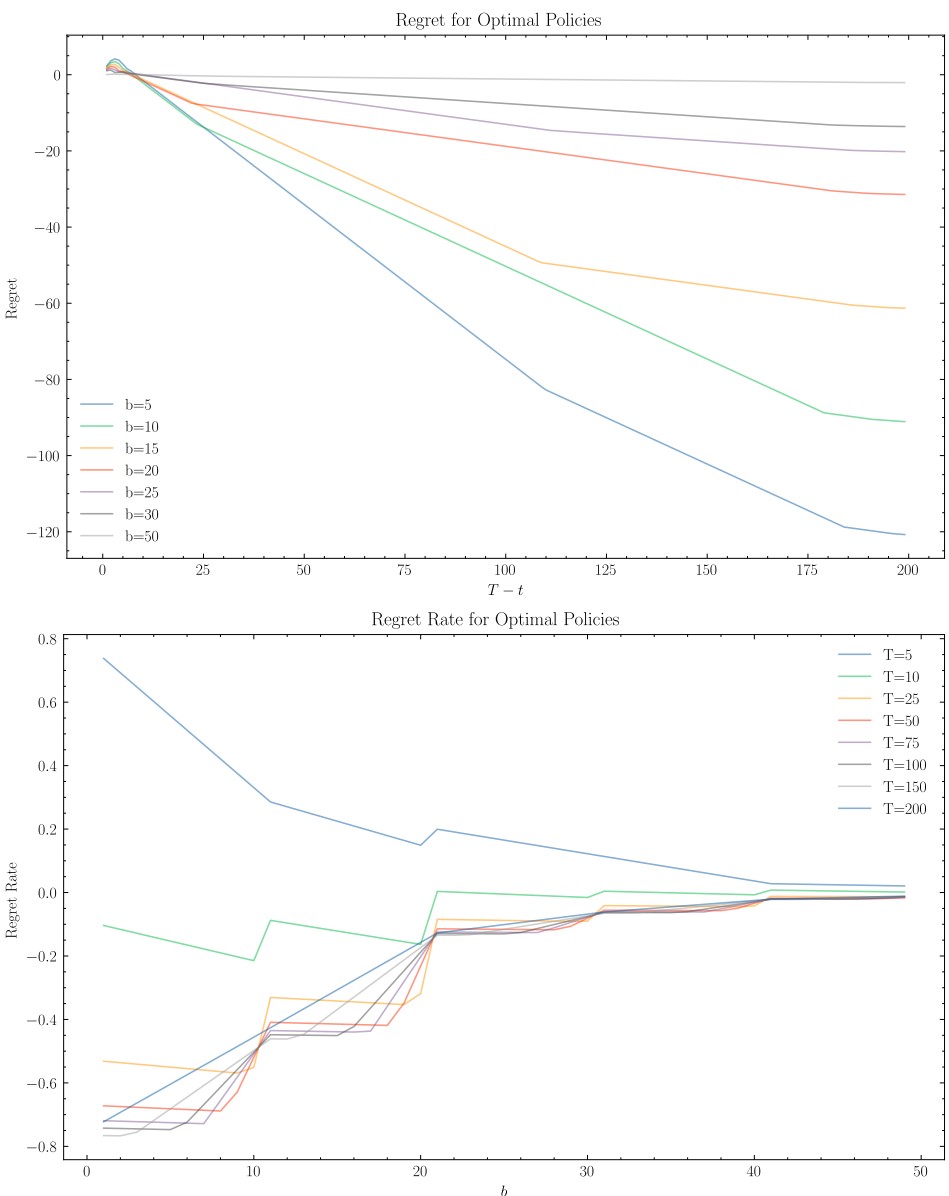

Figure 2: Comparison of Regret and Regret Rate for Different Time Horizons and Budgets

## D.2 High-Dimensional Action

We randomly generated a 10 action problem to simulate the risk aware behaviours of the agents. The problem has outcomes $\mathcal{Y} = \{Y_i\}_{i \in \{-20, -19, \dots 19, 20\}}$ ($|\mathcal{Y}| = 41$). The rewards of each outcome are simply its numerical order $R(Y_i) = i$. Each action outcome distribution has $|\text{supp}(p_a)| = 4$ picked at random from $\mathcal{Y}$, with a distribution uniformly sampled from the $4$−simplex. Note that given the structure chosen, all actions should sample outcomes that yield in expectation a mean reward of zero (but due to the variance of the sampling this is not the case for individual actions). The actions have expected (unclipped) rewards of $\mathbb{E}[R(Y_a)] = [-9.25,\ 8.11,\ 0.41,\ 3.37,\ -0.51,\ 2.01,\ -12.02,\ 0.10,\ 0.59,\ -1.55]$. As it can be seen, the optimal action is $a_2$ with $R^* = 8.11$. The plots for the regret and regret rate for different values of budgets and horizons. A very similar dynamic to the previous example emerges, as predicted by the results in Section 3. The regret is linear with time, since for lower budgets, higher time horizons force the agent to pick more conservative actions. Similarly, the regret rate is positive for very low

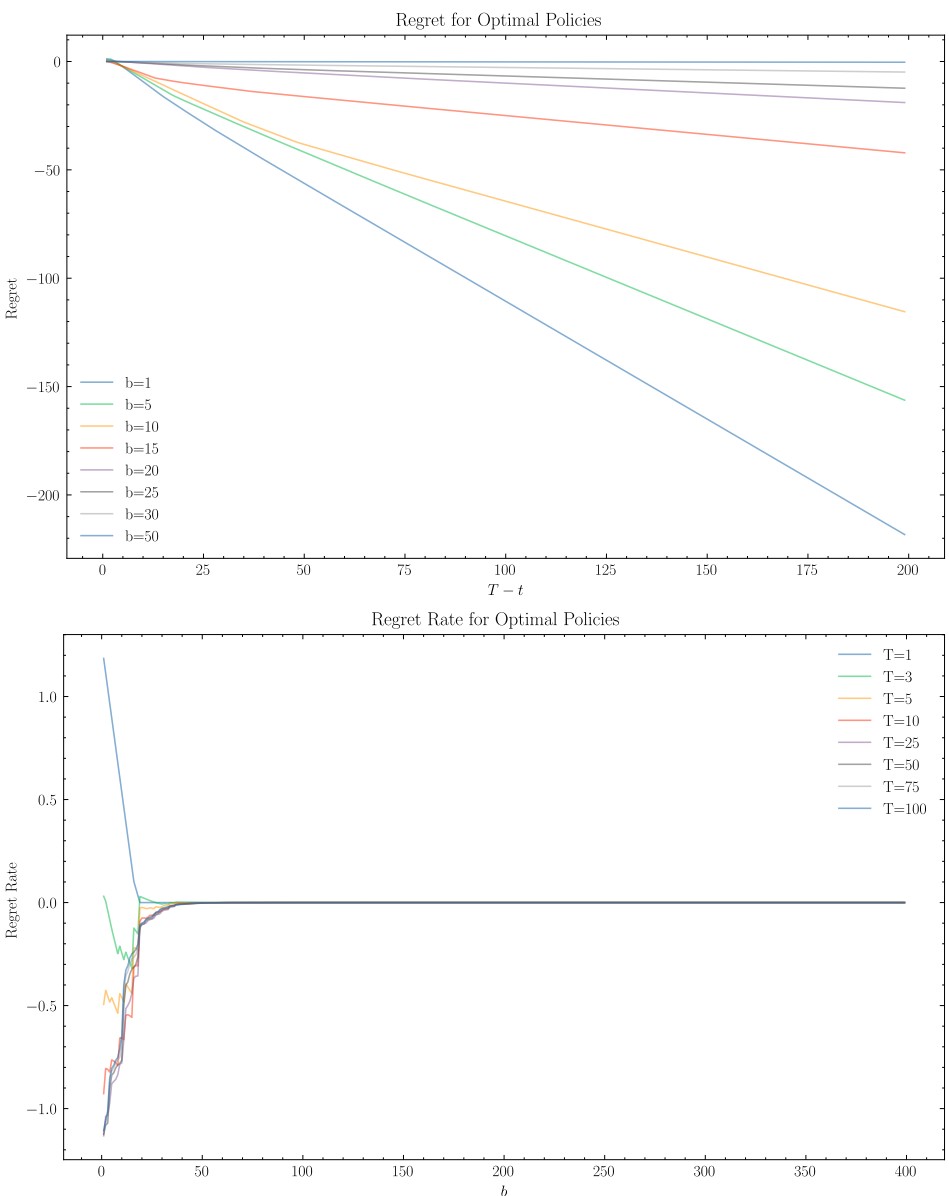

Figure 3: Comparison of Regret and Regret Rate for Different Time Horizons and Budgets for 10-action Bandit

horizons (indicating the agent exploits the limited liability), but quickly turns negative as the horizons increase, as the agent turns to actions with lower rewards but higher survival probability.

### D.3 Gambler's Problem

We generated an example of a gambling problem to further showcase the impact of the risk-seeking behaviour emerging in the agents. This problem has three outcomes, $\mathcal{Y} = \{Y_{bad}, Y_{safe}, Y_{good}\}$. The agent can choose between flipping two coins (two actions). The first coin induces $p_{a_1} = (0.5, 0, 0.5)$, with $R(Y_{bad}) = -10$ and $R(Y_{good}) = 10$. The second coin induces $p_{a_2} = (0, 1, 0)$, with $R(Y_{safe}) = 1$; the agent gets a fixed payoff of 1.. Without limited liability and survival awareness, there would be no reason to pick $a_1$, since $a_2$ is both the safe action in terms of survival probability and the optimal action with $R^* = 1$. In this case, there are no regions with negative regret since there is no possibility to be over-conservative; as soon as the limited liability property stops being advantageous, the agent

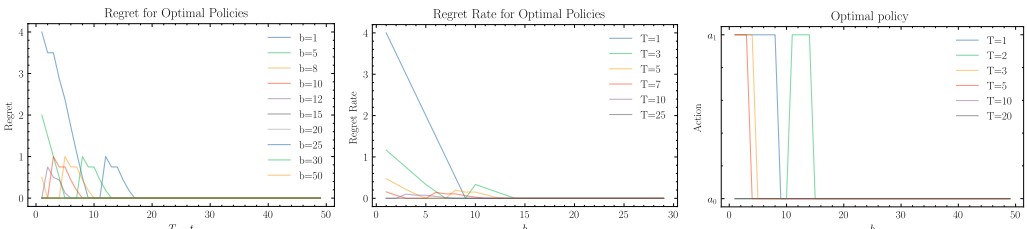

Figure 4: Comparison of Regret, Regret Rate and optimal policy for gambler's problem.

just falls back into picking the optimal (and safe) action. As a result, the agent exhibits what seems to be irrational behaviour: It chooses actions that are on average worse, and have higher uncertainty, simply since it estimates that given the survival constraint it will not be required to account for the negative payoffs in case of an undesired outcome.

## E  Learning Dynamics and Resource Constraints

The main analysis within this paper assumes that the agent possesses perfect knowledge of the outcome distributions $p_a$ for each action $a \in \mathcal{A}$. In many realistic scenarios, however, agents may operate with incomplete information, requiring them to learn about their environment over time. This appendix extends our framework to consider Bayesian agents that maintain beliefs over the parameters of these outcome distributions and may employ a Thompson sampling-like strategy to navigate the exploration-exploitation trade-off under survival constraints.

**Bayesian Model of Outcome Distributions**    Let the true outcome distribution for an action $a \in \mathcal{A}$ be parameterised by a vector $\theta_a \in \Omega_a$, such that $p_a(Y) = p(Y|\theta_a)$. The agent does not know the true parameters $\theta_a^*$. Instead, it maintains a posterior (or prior, before any observations) probability distribution $P(\theta_a)$ over the possible values of $\theta_a$. Let $\Theta = \{\theta_a\}_{a \in \mathcal{A}}$ represent the set of all such parameters for all actions, with a joint distribution $P(\Theta)$. We assume independence across action parameters, such that $P(\Theta) = \prod_{a \in \mathcal{A}} P(\theta_a)$. For instance, if outcomes $\mathcal{Y}$ are discrete, $p(Y|\theta_a)$ could be a categorical distribution, where $\theta_a$ is the vector of probabilities for each outcome given action $a$, and $P(\theta_a)$ could be a Dirichlet distribution.

**Learning and Posterior Update**    As the agent interacts with the environment by executing actions $a_t$ (chosen according to $\pi^*_{\hat{\Theta}}(b_t)$) and observing outcomes $Y_{a_t}$, it collects observed outcomes which can be used to update its beliefs about the parameters $\Theta$. For each action $a_k$ taken at time $k$ resulting in outcome $Y_{a_k}$, the posterior for its corresponding parameter $\theta_{a_k}$ is updated via Bayes' rule:

$$P(\theta_{a_k} \mid H_k, Y_{a_k}) \propto p(Y_{a_k}|\theta_{a_k})P(\theta_{a_k}|H_{k-1}),$$

where $H_k$ includes all action-outcome pairs up to time $k$. If the parameters for different actions are assumed independent, only the belief $P(\theta_{a_k})$ for the parameter corresponding to the action taken is updated. This updated posterior $P(\Theta|H_T)$ then serves as the basis for sampling outcome distributions in subsequent decision-making epochs.

**Implications of Bayesian Learning under Survival Pressure**    The introduction of uncertainty regarding $p_a$ and a Thompson sampling-like learning mechanism have deep implications for agent behaviour (and alignment) within the survival-constrained framework:

1. **Exploration and Survival:** Thompson sampling balances exploration (selecting actions with uncertain parameters to gain information) and exploitation (selecting actions believed to be optimal given current knowledge). Under survival pressure, exploring an action whose outcome distribution $p(Y|\hat{\theta}_a)$ (based on a pessimistic sample $\hat{\theta}_a$) suggests a high probability of depleting the budget $b_t$ might be deferred or avoided, even if its expected reward under $P(\theta_a)$ is high. Alternatively, an agent with a very low budget might attempt highly uncertain ("hopeful gambles") exploratory actions if all currently known "safe" options are not enough for survival.

2. **Belief-Dependent Risk Preferences:** The agent's effective risk preferences (behaving in a risk-neutral, survival-focused or risk-seeking manner) are not only a function of its budget $b_t$, horizon

$T$ and the (known) reward distributions $R(Y)$. These also depend on the current belief $P(\Theta)$. An action might seem optimal under the mean of $P(\theta_a)$, but a particular pessimistic sample $\hat{\theta}_a$ could render it too dangerous from a survival standpoint.

3. **Challenges for Alignment:** Asymmetries in information between the principal and the agent regarding the true outcome distributions $p_a$ can lead to more severe misalignment. The agent's prior $P(\Theta)$ could be misspecified by the principal, or early stochastic outcomes can lead to skewed posteriors. Then, the agent might learn an incorrect internal model of the environment, leading to behaviours that deviate from the principal's objectives, particularly if the learned model overestimates the probability of survival. The principal must now also consider how the agent learns and how to guide this learning process under survival pressure.

4. **Value of Information under Survival Constraints:** The agent implicitly faces a trade-off between actions that yield immediate high (clipped) reward $\tilde{R}$ and actions that provide valuable information for future decision-making by reducing uncertainty in $P(\Theta)$. The survival constraint critically shapes the perceived value of information: information gathering is of little use if the agent fails to survive to leverage it. This can lead to suboptimal behaviours based on incomplete knowledge if information-gathering actions are perceived as too risky for survival.

